# Evaluation of Resources and Environment Carrying Capacity Based on Support Pressure Coupling Mechanism: A Case Study of the Yangtze River Economic Belt

**DOI:** 10.3390/ijerph20010240

**Published:** 2022-12-23

**Authors:** Cheng Zhan, Mingjing Guo, Jinhua Cheng, Hongxia Peng

**Affiliations:** 1School of Economics and Management, China University of Geosciences, Wuhan 430074, China; 2School of Geography and Information Engineering, China University of Geosciences, Wuhan 430074, China

**Keywords:** RECC, resources and environmental support (RES), resources and environmental pressure (REP), Yangtze River Economic Belt (YREB), obstacle factors

## Abstract

Resource and environmental carrying capacity (RECC) is an important basis for achieving sustainable urban development, and analysis of the relationship between regional resources and human activities is of great significance for sustainable regional development. Taking the Yangtze River Economic Belt (YREB) as the study area, this study establishes a framework for analyzing RECC based on the resource and environmental support capacity (RES) and the pressure on the resource and environment (REP), calculates the RES and REP of 110 cities in the YREB from 2009 to 2018, and analyzes the main constraints on RECC. The results show that (1) there are inter-regional imbalances in RECC within the study area, with cities that are more economically developed or at a higher administrative level usually having more severe problems with RECC. (2) The RES and REP indices of cities in the YREB show an overall increasing trend, but the relative growth rates of the RES and REP indices of cities at different levels differ. (3) The built-up area, green space in built-up areas, total gas supply, and length of sewage pipes are hindering factors for most cities to improve their RES. This study contributes to a comprehensive understanding of the current situation and changing trends of RECC in the YREB and can provide a reference for decision-making on sustainable development of the region’s large river basin.

## 1. Introduction

### 1.1. Background

The Yangtze River Economic Belt (YREB) is the most economically developed and competitive economic belt in China, the most populous and largest river basin economic belt in the world, and an inland economic belt with global influence. In recent years, China’s rapid socio-economic development and accelerating urbanization have increased the rate of urban consumption of natural and social resources, which in turn has led to serious urban environmental degradation and resource depletion, creating a vicious cycle. The ecological space of the city is heavily encroached upon by production space and living space [1]. Problems such as haze pollution [2], water shortages [3,4,5,6], human–land conflict [7,8], loss of biodiversity [9], destruction of ecosystems [9], traffic congestion, and regional development gaps [10,11,12] are becoming increasingly serious [13,14,15,16,17,18]. Improving the urban RECC is necessary not only to improve the economic development of cities but also to accelerate the urbanization process and improve the quality of urban development. Given that sustainable development has become the overriding theme of urban development, urban RECC has become an important topic of concern for the Chinese government [19]. China’s 14th Five-Year Plan (2020–2025) states that China’s goal is “to achieve a significant green transformation in production and lifestyle, a more rational allocation of energy resources and a significant increase in their efficiency, a continuous reduction in total emissions of major pollutants, a continuous improvement in the ecological environment, a stronger ecological security barrier, and a significant improvement in the urban and rural living environment”. With the rapid development of urbanization and industrialization in China, the contradiction between socio-economic development and population, resources, and the environment is becoming increasingly prominent and has become a major bottleneck for sustainable regional development. The concept of carrying capacity describes the relationship between resources, the environment, and human activities to ensure sustainability. However, the relationship between these elements still requires a profound explanation.

### 1.2. Literature Review

Resource environmental carrying capacity (RECC) is a scientific concept that measures the relationship between human socioeconomic activities and the natural environment and is an important tool for measuring and managing sustainable human development. Relative stability of the RECC is the precondition for long-term rapid development [20]. Carrying capacity is defined as the ability of a carrier to support a ‘load-bearing object’. Carrying capacity theory has its origins in the fields of demography, applied ecology, and population biology. In the 200 years since its inception, the theory has made achievements and developed considerably; its applications have expanded, and it has evolved to address practical issues of human economic and social development [21,22,23]. The concepts of resource carrying capacity and environmental carrying capacity were originally derived at different times and under different circumstances and are heavily applied in the field of resources and the environment. The carrier is generally a natural system, and the main purpose is to study its ability to support human economic and social activities as a carrying object. Since the introduction of the concept of RECC, with the deepening of human research and the increasing complexity of resource and environmental problems, the connotation of the carrier has been enriched and developed from the initial development of single elements, such as land resources and water resources, to research of comprehensive elements, such as natural resources, resource environment, and ecological environment, as carriers. Resource carrying capacity includes mainly land resources, water resources, and mineral resources carrying capacity. The environmental carrying capacity includes atmospheric environment, water environment, and soil environment carrying capacity. In addition, RECC is constrained and influenced by relevant factors, such as level of economic development, use of science and technology, social and cultural habits, and the degree of regional openness [24].

RECC studies are analyzed from a comprehensive perspective, mainly by constructing a systematic framework containing each carrying capacity element and analyzing and studying each carrying capacity element according to a unified characterization value [25]. Regional RECC is particularly important in carrying capacity studies; most studies first analyze the type of carrying capacity, determine the indicators affecting the carrying capacity of the resource, establish an evaluation indicator system, determine the indicator weights, and use various evaluation methods to complete the evaluation. Establishing the evaluation index system is the core content of conducting RECC research [26]. The RECC evaluation index system should follow the basic principles of scientificity, representativeness, regionality, and completeness and combine the characteristics of regional resources and environment as well as the purpose of the study to select the elements and index factors affecting regional RECC. In constructing the indicator system, researchers rely on two common processes: the first process assumes that resource, environmental, economic, social, and other criteria are additive; the other constructs the stress state response as a system layer [27,28,29,30]. Several models can be used in the assessment, such as the ecological footprint approach, the water footprint approach, the primary asset accounts approach, the carbon and oxygen balance approach, the state space approach, and system dynamics models, each with a different focus. For example, ecological footprint models compare the supply and demand relationship between the demand for ecological resources from human development and the services provided by ecological resources in the area and assess the impact of human activities on natural systems to guide more efficient and rational allocation of resources. A system dynamics model [31] can quantify the intrinsic relationships between the structure and function of various complex systems and quantitatively analyze various characteristics, making them suitable for objective, long-term studies of dynamic trends. TOPSIS models have advantages of horizontal and vertical comparative analysis and simplicity of calculation. In recent years, research on carrying capacity has led to recognition that environmental disturbances caused by human activities indicate the fragility and variability of stability, the integrity of environmental systems, and their ability to resist external disturbances and can easily lead to qualitative changes or mutations in the structure and function of environmental systems. These findings have facilitated application of mutation sequence models in carrying capacity studies, which to some extent reflects the discontinuous and potentially mutational character of RECC.

Currently, RECC research in China is an important topic. For example, Jiang Hao et al., (2016) conducted a dynamic analysis of Chengdu’s comprehensive urban carrying capacity from 2008 to 2014 from the integrated dimensions of natural resources, socio-economics, and ecological environment, where the ecological environment is a necessary factor in determining the comprehensive urban carrying capacity of Chengdu, and the influence of natural resources on the comprehensive urban carrying capacity of Chengdu was smaller [32]. Duan Peli et al., (2019) studied the coupling and coordination degree of development intensity and RECC of five major urban agglomerations in China in 2015. The overall coupling degree of development intensity and RECC of urban agglomerations is low [33]. Zhang, Ningning, et al., (2019) constructed a four-dimensional evaluation index system of “quantity-quality-domain-flow” to comprehensively evaluate the water resources carrying capacity of 61 prefecture-level cities in the Yellow River Basin in 2015, and the results showed that the percentage of cities with overloaded water resources carrying capacity reached 77% [34]. Based on the variable-weight TOPSIS model, Tian Pei et al., (2019) determined the water resources carrying capacity of the YREB from 2015 to 2017, and the results show that the water resources carrying capacity improved but is still low and has potential for improvement [35].

A number of limitations exist in the current stage of research on RECC.

(1)In existing studies, most scholars often divide the level of RECC index (usually a single quantitative value between 0 and 1) at equal intervals or directly use the grading standards of previous literature, which can only indicate the change in the intensity of RECC and cannot indicate the carrying state of RECC (i.e., overload state or surplus state). These subjective evaluation standards not only weaken the explanatory power and persuasiveness of the evaluation results to a certain extent but also cannot determine whether the pressure of human activities on resources and environment exceeds the carrying capacity of the resources and environment, which is not conducive to an accurate understanding of the sustainable state of RECC.(2)In existing studies [36,37], although RECC systems are usually divided into resource, environmental, and human activity layers, they are still eventually calculated as a whole, and the final results are unable to characterize the attributes of two relatively independent systems. In other words, the support capacity of resources and environment and the overall pressure of human activities on these systems are calculated without considering the coupling relationship between the two systems. It is not conducive to understanding the current state of these two systems or to further understanding the impact of resource and environmental support activities and human activities on these systems.(3)At present, there are few studies of the YREB RECC at the city level. Relevant studies have focused on specific regions [38,39,40]. However, the lack of research on RECC at the city level is not conducive to establishment of a uniform early warning mechanism for RECC at the national level.

By combining human activities with resources and environment, this paper constructs an RECC evaluation system including support and pressure factors affecting bearing capacity, which complements the existing research. The support index represents the potential of natural resources and the social public resources created by human beings to promote development. The pressure index represents the economic and social activities of human beings served by the support system. Therefore, in this paper, the resource and environment carrying capacity index (RECC), which is a single quantitative value in most other studies [41], can be expressed as the ratio of the support index and pressure index. Then, this paper determines the critical value in the overload (surplus) area according to the coupling mechanism between the support side and the pressure side, which can help to refine the RECC partition. In the field of social science, there are roughly two methods to measure coupling between two or more systems: coupling degree model and coupling curve graph. The coupling degree model can express the coupling degree of support and pressure but cannot comprehensively reflect the state to RECC, so the critical value of RECC partition cannot be obtained. The coupling curve chart can express different coupling states and itself includes consideration of the critical value, and it can be transformed into an expression mode through appropriate mathematical transformation to meet the needs of the research object. Therefore, this paper constructs a coupling curve graph containing two variables, namely the support side and the pressure side, to express the different coupling states of RECC and obtains the critical value of RECC partition through mathematical transformation of the coupling curve. Then, we can discuss the status of RECC according to the ratio and support pressure coupling curve. It not only makes the evaluation results of RECC more convincing but also better reflects whether RCCC has overload/surplus. With this improved RECC index, we can scientifically determine the strength of urban resources and environmental carrying capacity, which helps us to understand the changes in the RECC and urbanization process of the YREB cities in China and establish an RECC early warning mechanism.

## 2. Data and Methods

### 2.1. Study Area

The YREB is an economic circle near the Yangtze River that includes two municipalities (Shanghai and Chongqing) and nine provinces (Jiangsu, Zhejiang, Anhui, Jiangxi, Hubei, Hunan, Sichuan, Yunnan, and Guizhou) (Figure 1). The YREB spans three regions in China, east and west, and has unique advantages and great development potential. The YREB accounts for 21.4% of China’s land area in terms of land resources and 42.9% of China’s population and contributed 46.5% of China’s GDP in 2020 (China Statistical Yearbook 2021). In recent years, the coercive effects of rapid urbanization on the YREB ecosystem have become increasingly prominent, seriously threatening the sustainable development process of the YREB. Therefore, scientific evaluation of RECC in each city of the YREB is important for formulation of sustainable development policies in the YREB in general and in specific cities in particular. In this paper, 110 cities in the YREB, including municipalities directly under the central government, provincial capitals, and prefecture-level cities, are studied.

### 2.2. Data Sources

Water resource data were obtained from the cities’ water resources bulletins from 2009 to 2018 (Urban Water and Lakes Agency). Other natural resource data, infrastructure data, and economic-development-related data were obtained from the cities’ statistical yearbooks from 2009 to 2018 (Urban Statistics Agency). The environmental quality data are from the cities’ annual environmental quality bulletin (Urban Ecological and Environmental Protection Agency). Social-security-related data and environmental pollution data were obtained from the cities’ annual National Economic and Social Development Statistical Bulletin from 2009 to 2018 (city government).

### 2.3. Research Methods

#### 2.3.1. Construction of the RECC Evaluation Index System

Regional resource and environmental carrying capacity should consider how resources, the environment, and human activities interact with each other. RECC can be examined from two perspectives: the supporting capacity of resource and environmental systems and the pollution and damage to resources and the environment caused by socio-economic activities. Within the RECC system, the support index is divided into two subsystems: the resource and environmental systems. Human social activities can be broadly summarized into three categories: economic development, social state, and environmental pollution. After combining the available data from 110 cities, 10 RES indicators and 10 REP indicators were identified (Table 1). Based on the RECC evaluation model, this paper attempts to establish a governance assessment index system for resource and environmental stress tolerance based on the basic principles of scientific, systematic, regional, hierarchical, openness, and dynamism with reference to the literature on research index systems and quantification and availability of data.

#### 2.3.2. Standardization of index

In this paper, the extreme value method is used to standardize the data and ensure that the standardized data are between 0 and 1. The specific method is as follows.
(1)$${x^{\prime }}_{ij}=\frac{x_{ij}-minx_{ij}}{maxx_{ij}-minx_{ij}}\;$$

In Equation (1), $x_{ij}$ and ${x^{\prime }}_{ij}$ represent the original value and standard value of the index, respectively.

#### 2.3.3. Weight Measurement

Evaluation of RECC must assign weights to each index. Through collation and comparison of the literature and analysis, this paper uses the analytic hierarchy process (AHP) variation coefficient method (VCM) method to determine the weights. The AHP has strong subjectivity, so most studies will use it in combination with some objective weight calculation method [42,43]. In this paper, the variation coefficient method (VCM) is used to determine the objective weight, which can reduce workload and overcome the adverse effects of abnormal values [44]. The AHP is used to determine the subjective weights of indicators, including establishing the comparison matrix, calculating the weight vector, verifying the consistency of the judgment matrix, and determining the indicator weights. The objective weights of indicators are determined using the CV method. Since the support system and the pressure system apply the same weighting method, the RES index is used as an example to describe the process of standardization and calculation of indicator weights. The specific calculation is as follows:(2)$$W_{i}^{S^{\prime }}=\frac{\sigma_{i}}{{\bar{x}}_{i}},i=1,2,3\cdots ,n\;$$

In Equation (2), $\sigma_{i}$ is the standard deviation of the ith index, ${\bar{x}}_{i}$ is the average value of the ith index, and $W_{i}^{S^{\prime }}$ is the coefficient of variation of the ith index of RES.

The AHP-CV method was used to determine the combined weights of the evaluation indicators [45].
(3)$$W_{i}^{S}=\beta W_{i}^{S^{\prime }}+\left(1-\beta \right)W_{i}^{S^{\prime \prime }}$$

In Equation (3), *β* is the preference coefficient, 0 ≤ *β* ≤ 1. After consulting relevant experts, the coefficient *β* was set to 0.5. $W_{i}^{S^{\prime }}$ is the weight value of index I calculated by the EWM, and $W_{i}^{S^{\prime \prime }}$ is the weight value of the ith index of the RES calculated by AHP.

#### 2.3.4. Calculation of the RECC Index

Based on the normalized values of each indicator and the corresponding weights, the RES index and REP index are obtained using the linear weighting method. The calculation method is as follows [46].
(4)$$S_{i}=\overset{n}{\underset{j=1}{\sum }}{S^{\prime }}_{ij}W_{j}^{S}\;$$
(5)$$P_{i}=\overset{n}{\underset{j=1}{\sum }}P_{ij}^{\prime }W_{j}^{P}\;$$

Then, the RECC index for a city is equal to the ratio of RES to REP for that region. The results of the calculation are as follows.
(6)$$\mathrm{RECC}=\frac{S_{i}}{P_{i}}$$

Theoretically, according to the RES and REP indices, the RECC size can be divided into three cases, RECC < 1, RECC = 1, and RECC > 1: when RES = REP, i.e., RECC = 1, RECC is adequate for the stressful activities exerted by human beings. When RES > REP, i.e., RECC > 1, the capacity of the resource-environment system to support the population is stronger than the pressure exerted on the system by human social activities. When the pressure exerted on the support side is within the limits, there is a certain amount of residual capacity. When RES < REP, i.e., RECC < 1, the support capacity is less than the pressure exerted by human activities, and the regional RECC is overloaded. Considering the large differences in geographical location, resource endowment, environmental capacity, level of economic development, level of social development, and human activities, the RECC status of each city can be classified to guide human activities in a more targeted manner.

In this paper, after referring to the literature, RECC states are classified into five types, and the classification criteria are based mainly on functions y=x, $y=x^{2}$ and $y=\sqrt{x}$ [47]. The specific classification criteria and schematic diagrams are shown in Table 2 and Figure 2.

In addition, RECC states can be divided into four types according to the size range of RES and REP indexes (Table 3).

The “L-H” RECC status indicates that the city has a low RES and a high REP. In the “L-H” state, although a city can achieve great economic and social development in the short term, it is not conducive to sustainable development in the long term. The best way to develop a city in an ‘L-H’ state is to gradually move to an ‘H-H’ state.

The “H-H” RECC status indicates a city with a high RES and REP. A city with the “H-H” status has not only a high level of economic and social development but also good resources and environmental factors to support it and strong development momentum and development prospects. When the REP of cities with the “H-H” status gradually decreases, they will gradually shift to the “L-H” status. Each city should adjust its development status appropriately according to its own conditions and economic development level.

The “L-L” RECC status indicates that the city’s RES and REP are both low. A city with the “L-L” status not only has a low level of economic and social development but also lacks the resources and environmental factors to support economic and social development, and the city’s development prospects are constrained.

The “H-L” RECC status indicates that the city has a high RES and a low REP. A city with the “H-L” status has a low level of economic and social development but has a strong resource and environmental endowment. In addition to lagging development, sacrifice of economic development to protect ecological and environmental resources is a reason for the low level of economic and social development in these cities. However, this state of development is not sustainable, and cities with the “L-L” state should make full use of their resource and environmental endowments to actively explore a green and ecologically sustainable development path.

#### 2.3.5. Analysis of Obstacles

A barrier degree model was introduced to explore the main barrier factors of the YREB RECC support system to effectively diagnose the pathology of the shortcomings of the YREB resource environment and to propose targeted enhancement strategies with a model structure [48].
(7)$$M_{ij}=\frac{d_{ij}\times W_{ij}}{\sum_{j=1}^{n}d_{ij}\times W_{ij}}\times 100\%\;$$
(8)$$d_{ij}=1-x_{ij}^{\prime }\;$$

$M_{ij}$ represents the obstacle degree of a single index to RES, which is the obstacle degree of indices J in city I; $d_{ij}$ is the deviation degree of the index, which represents the gap between the single index and the maximum target and is set as the gap between the standardized value of the index and 1; $W_{ij}$ represents the contribution of the index to the overall goal, which is the weight value of J indicators in city I; $x_{ij}^{\prime }$ is the standard value of the j-th index in criterion I; and n is the number of indicators.

## 3. Results

### 3.1. RES and REP Analysis

The RES and REP indices for the 110 cities in the YREB can be calculated using the RES and REP models. Both the RES and REP indices show significant spatial variation within the YREB (Figure 3 and Figure 4), while there are differences in the RES and REP indices for different cities at the same administrative level (Figure 5 and Figure 6).

(1)Whether in 2009 (Figure 3a) or 2018 (Figure 3b), the two municipalities, Shanghai and Chongqing, have much higher RE indices than other cities and are the peak cities in the RES index in the eastern and western regions of the YREB, respectively. Furthermore, the RES index tends to decrease with the distance from these two cities to the periphery, while the provincial capitals generally have the next highest values of the RES index.(2)Unlike the change in the RES index, the trend of the REP index across cities in the YREB from 2009 (Figure 4a) to 2018 ( Figure 4b) is complex and does not have a pronounced trend in spatial terms. On the other hand, when analyzed by city administrative level, the two municipalities, Shanghai and Chongqing, remain the peak cities for the REP index in the eastern and western regions of the YREB, respectively, and provincial capitals are also generally the sub-peak cities for the REP index within the YREB.(3)Figure 5 shows that the RES index differs significantly between different cities within the same administrative level: the higher the administrative level of a city, the larger the RES index value usually is. Moreover, the higher the administrative level of a city is, the smaller the difference in the RES index between different cities within that administrative level. Compared to 2009, by 2018, the RES indices of both Shanghai and Chongqing municipalities had increased, but Chongqing’s RES index increased significantly faster than Shanghai’s; therefore, the gap in RES indices between the two increased significantly. The overall level of RES index for provincial capitals changed less, but the minimum, mean, and median increased and the extreme difference decreased, indicating that the gap in the RES index within provincial capitals is decreasing. Similar to the situation in provincial capitals, the minimum and extreme differences of the RES index for prefecture-level cities have also decreased, and the gap in the RES index between prefecture-level cities has decreased. As the RES index mainly reflects the city’s resource security capacity and environmental governance level, the higher the administrative level of a city is, the higher the resource security capacity and environmental governance level of the city. This may be related to the city’s higher level of economic and social development, so it has more sufficient financial resources to invest in related areas. On the other hand, the changes in the RES index for provincial capitals and prefecture-level cities suggest that some tail-end cities have made efforts to improve their resource security capacity and environmental governance levels and have achieved some desired results.(4)Figure 6 shows that, similar to the case of the RES index, the higher the administrative level of a city is, the greater its REP index. In contrast to the change in the RES index, the difference in the REP index between the two municipalities decreased by 2018 compared to 2009. In contrast, the minimum, median, mean, and extreme differences of the REP index for provincial and prefecture-level cities all increased. As the REP index reflects mainly the level of economic development, social security capacity, and environmental pollution of cities, the levels of economic and social development and environmental pollution of provincial capitals and prefecture-level cities are all increasing. However, there is a large difference in the development rate between different cities at the same administrative level, with the head city within the level improving more significantly and the gap in development level between cities increasing further.

### 3.2. RECC Analysis

According to the RECC evaluation model, RECC indicators can be further calculated through the RES index and REP index to obtain RECC indicators. The breakdown of RECC status and city types for the 110 cities in the YREB in 2009 and 2018 is shown in Figure 7.

As shown in Figure 7, Figure 7a,d shows the coupling of RES index and REP index in different cities in 2009 and 2018. Figure 7b,e shows the types of RECCs in different cities in 2009 and 2018, while Figure 7c,f shows their spatial distribution.

(1)Figure 7a–c shows that. in 2009, both municipalities directly under the central government were in RECC overload, with Shanghai in the high-level overload region and Chongqing in the low-level-overload region. Three provincial capital cities, Hangzhou, Wuhan, and Chengdu, had RECC in the low-level overload area. Among the prefecture-level cities, only Suzhou had RECC overload and was in the low-level overload area, while the RECC of other cities had not yet reached overload status, among which 59 cities are in the low-level surplus area and 45 cities are in the high-level surplus area. Meanwhile, all cities are L-L cities, except Shanghai and Chongqing, which are H-H cities.(2)Figure 7e,f shows that, by 2018, among the municipalities directly under the central government, Shanghai’s RECC was eased from a high-level overload area to a low-level overload area, with a further trend towards a low-level surplus area, while Chongqing’s RECC improved from a low-level overload area to a low-level surplus area. Among the provincial capitals, Wuhan and Chengdu’s RECCs were still in the low-level overload area, while Hangzhou’s RECC had improved from a low-level overload area to a low-level surplus area. Among the prefecture-level cities, the RECC of Wuxi also changed from a low-level surplus area to a low-level overload area, except for Suzhou, where the RECC deteriorated from a low-level overload area to a high-level overload area. The RECCs of the remaining cities had not yet reached overload status, with 60 cities in the low-level surplus area and 44 cities in the high-level surplus area. Shanghai and Chongqing were H-H cities, in addition to Chengdu and Suzhou, which had transformed from L-L to L-H cities, while the remaining cities were still L-L cities. This indicates that most of the YREB cities still had rich resource potential and environmental capacity and more room for future development.(3)By comparing the changes in the RECC classification and status of the YREB’s 110 cities as of 2009 and 2018, it can be found that the improvement in the RECC status of Shanghai and Hangzhou is mainly due to the decrease in their REP indices, indicating that they have effectively controlled their environmental pollution levels. The improvement in the RECC status of Chongqing is due to the combination of a significant increase in the RES index and a decrease in the REP index, indicating that its resource security capacity and environmental management level have been significantly improved and its environmental pollution situation has been improved. The deterioration in the RECC status of Wuxi and Suzhou is caused by the decrease in the RES index and the significant increase in the REP index, which indicates that they have not only under-invested in their resource security capacity and environmental governance level but also experienced deterioration in their environmental pollution situation. These cities must take necessary measures and increase their financial investment to improve this disadvantage.

Due to the dynamic economic and social development of the YREB, the interaction between human activities and resource and environmental systems in the area is frequent and complex. Factors including changes in population, changes in industrial structure, intensity of urban construction, type and size of land use, paving of urban roads and other infrastructure development, improvements in resource use efficiency, and significant improvements in environmental technologies inevitably have a significant impact on the city’s RES and REP indices, so the RES and REP indices of other cities are also in a dynamic state of change. However, as these cities are at a relatively low level of development, their own resource and environmental potential can still support their economic and social development consumption; therefore, their RECCs have not yet reached an overload state. These cities must pay attention to improving their resource security capacity and environmental protection in the future development process and follow the path of sustainable development.

Notably, while cities other than Shanghai, Chongqing, Chengdu, and Suzhou are all L-L cities, some cities, such as Wuhan and Wuxi, are in a critical position and are trending towards L-H cities. If these cities fail to pay attention to improving resource efficiency and environmental protection efforts while developing their economies and shift from pursuing high economic growth to a model of high-quality economic development, their economic and social development will not be sustainable. Their governments must promptly adjust their development paths, reform their economic development models, and follow a sustainable development path to achieve simultaneous growth in the RES and REP indices and eventually develop towards H-H cities.

### 3.3. Analysis of Limiting Factors

By analyzing the current barriers to RES indicators in the RES system and identifying the shortcomings of RES indicators, targeted policies and recommendations can be made for the future development of cities. Based on the barrier degree calculation method, it is possible to derive the barrier degree of each indicator supporting large cities in 2018. Figure 8 illustrates the spatial distribution of RES indicator barrier degrees and barrier degrees for YREB cities.

Figure 8 shows the obstacle factors of different cities and their spatial distribution.

(1)Figure 8 shows that factors such as built-up area (S3) (Figure 8c), green space of built-up area (S4) (Figure 8d), total gas supply (S6) (Figure 8f), and length of sewerage pipes (S8) (Figure 8h) are common barriers to RECC in most YREB cities. These barriers have a significant impact on cities in the YREB other than Shanghai in terms of enhancing their RECC capacity. These barriers mainly fall under the social resource factor in the resource system and the environmental governance factor in the environmental system. Thus, lack of investment in social resources and environmental governance is a common barrier for most YREB cities to improve their support capacity. These cities can be classified as ‘social resource and environmental governance barrier’ cities. They face not only insufficient investment in social resources but also low levels of environmental governance and need to accelerate improvement in infrastructure, such as built-up area, built-up green space, total gas supply, and length of sewerage pipes, to improve the city’s support capacity.(2)Figure 8a shows that total water resources (S1) (Figure 8a) is the main obstacle to improving the supporting capacity of cities such as Shanghai, Suzhou, Wuxi, Nanjing, Wuhan, and Zigong. The main reason for this issue is that these cities have insufficient resource endowments of their own, and it is difficult to improve them effectively in the short term. They can alleviate water stress by strengthening water conservation facilities and improving water use efficiency. In addition, shows that grain sowing area (S2) (Figure 8b), built-up area (S3) (Figure 8c), and road mileage (S5) (Figure 8e) are also main barriers to improving support capacity in Shanghai. Shanghai differs significantly from the other YREB cities in that it has a smaller land area (85th) and an extremely high urbanization rate (over 90%), making it less well endowed in terms of natural resources and extremely rich in social resources. Furthermore, Shanghai does not have significant barriers to its environmental system due to its relatively good revenues, which enable it to invest heavily in environmental pollution control projects. These cities can be classified as ‘natural resource barrier’ cities. As natural resources are naturally endowed and cannot be enhanced in a short period of time, these barriers can be improved by increasing efficiency of resource use and enhancing rational distribution of regional resources.(3)Obstacle factors, ratio of good air quality (S7) (Figure 8g), ratio of centrally treated wastewater within sewage works (S9) (Figure 8i) and ratio of domestic refuse disposal (S10) (Figure 8j) have less than 10% impact on all cities, so they are not the main obstacle factors.

## 4. Discussion

In this paper, the RECC system is divided into two parts: the RES index, which reflects the resource security capacity and the environmental condition and governance level; and the REP index, which reflects the pressure of economic activities, social security, and environmental pollution on the resource and environmental system. The indicators are weighted: the entropy weighting method is used to determine the weight coefficients of each indicator and the linear weighting method is used to calculate the RES index and REP index. In this paper, the RECC index is the ratio of RES index and REP index rather than the simple linear weighted sum (usually between 0 and 1). In addition, combined with the coupling curve, it can not only calculate the size of the RECC but also judge the status of the RECC (overload/surplus) so as to more accurately formulate targeted development strategies for cities with different RECC statuses.

This paper finds that, among the 110 cities in the YREB between 2009 and 2018, the higher the administrative level of a city is, the higher the RES index and REP index of that city. Considering that, in China, the higher the administrative level of a city is, the better its level of economic and social development in general, it can, therefore, be concluded that, the more developed a city is, the more supportive it is of its resource and environmental system and the more pressure human activities place on it. This is similar to the findings of the study by Zhang Fei et al. on RECC in large cities in China. In that study, they found that Chinese megacities face greater pressures on the resource environment. The study by Shang Yongmin et al. on the RECC of cities in the YREB also shows that the economically developed eastern coastal areas face higher resource and environmental pressures than the western inland areas. Other studies have also found that the more developed cities, such as Shanghai and Guiyang, perform better between coordination of economy and RECC [49,50]. Some teams found that the RECC-related indicators all showed an increasing trend at the provincial level [51]. Han’s team approach research also shows that the coordination between urbanization and the water resources system has evolved from a serious imbalance to a good coordination [52].

This paper differs from other existing studies in that it finds that the gap in the RES index between the municipalities directly under the central government, Shanghai and Chongqing, is becoming wider, while the gap in the RES index between different provincial capitals or different prefecture-level cities is decreasing. In contrast to the change in the RES index, the gap in the REP index between the municipalities directly under the central government, Shanghai and Chongqing, is decreasing, while the gap in the RES index between provincial capitals and within prefecture-level cities is increasing. The change in the RECC index is more complex as it is calculated by combining the RES index and the REP index. According to the research in this paper, there is a large difference between the RECC of a municipality or provincial capital city in the YREB and its neighboring prefecture-level cities. These cities are the central cities of the surrounding region, with higher levels of economic and social development and greater pressure on the resources and environment caused by human activities, so they are mostly in the resource and environmental overload zone. Moreover, cities in the eastern part of the YREB are under greater pressure on RECC than cities in the west. This phenomenon reflects the reality of China’s economic and social development, where urban development is being severely challenged by spatial imbalance, with each province often having only one development center, the provincial capital city, which can absorb the province’s resources for priority development. Furthermore, due to their location, cities in the east usually have a better level of economic and social development than cities in the west, and the combination of these factors creates uneven levels of urban development between and within provinces. Therefore, the government has realized intra-provincial economic linkages to accelerate development of prefecture-level cities, thereby easing the pressure on resources and the environment in the central cities. In addition, the RECC statuses of Shanghai, Chongqing, and Hangzhou all improved, while Suzhou and Wuxi experienced a deterioration in their RECC status.

Using a barrier degree model, this paper finds that built-up area, built-up green space, total gas supply, and length of sewerage pipes are the main barriers faced by most cities in the YREB, except Shanghai. Cities must improve urban infrastructure, optimize social resources, and increase investment in environmental management. Shanghai, on the other hand, due to its smaller area and higher level of development, is constrained mainly by the total water resources, sown area, built-up area, and road mileage and needs to focus on improving the green efficiency of resources [53].

Combined with the above analysis, the following policy suggestions are formulated to provide a relevant basis for the sustainable development of the YREB.

For municipal authorities and provincial capitals, their economic development level has reached a high level, so the relationship between human activities and resources and environmental systems in these cities is extremely tense. These locations should implement strict water resource management and rational land use planning in the future, and large-scale resource redeployment projects across regions, such as south–north water transfer and west–east gas transmission, still need to be built. In addition, these cities should increase the supply of social resources, such as urban green space and urban road areas. To solve the problem of resource scarcity, cities must change their production methods and lifestyles to improve the efficiency of resource use. Advanced cities should improve green innovation capabilities [54], develop waste management and green technologies, such as domestic garbage and water resources recycling technology [55], and improve green capabilities to achieve sustainable operation [56].

For the prefecture-level cities, their economic development is not good enough, which strongly necessitates economic development. Therefore, they have to make development plans in advance, prioritize development of resource-saving and environment-friendly industries, and focus on development of the circular economy. The government should strengthen investment in environmental management and improve coverage of technologies such as water purification technology, industrial sulfur dioxide treatment technology, industrial wastewater treatment technology, and domestic waste treatment technology to effectively reduce sewage discharge and waste generation. The city should pay attention to reducing the impact on and damage to existing resources and the environment while developing economically and actively exploring the path of sustainable development.

This paper evaluates only some of the cities’ resources, environment, and economic and social activities and does not address trade and other mobility factors between cities and regions. In addition, arable land area, forest area, food production, energy consumption, and some other factors were not considered due to problems in data collection. The driving forces behind changes in RECC in cities have not been studied and need to be assessed in future studies.

## 5. Conclusions

Based on the RES and REP coupling mechanism, this paper examines the RECC situation and trends of 110 cities in the YREB from 2009 to 2018. In addition, the main obstacles to the cities of the YREB in 2018 to improving their support capacity are analyzed. The main conclusions are as follows:(1)Within the YREB, there are regional imbalances in RECC such that, the more economically and socially developed a city is, or the higher its administrative level, the more serious its RECC problems are. This is reflected mainly in the fact that municipalities and provincial capitals are generally the most heavily loaded cities in the region in terms of RECC, and most of them are already in a state of overload, with the pressure of their human and social activities exceeding the capacity of the local resources and environmental services.(2)Both the RES and REP indices of cities in the YREB show an overall increasing trend, but the relative growth rates of the RES and REP indices vary depending on the administrative level of the city. Among them, the difference between RES indices of different municipalities directly under the central government increases and the difference between REP indices decreases; the difference between RES indices of different provincial capitals or different prefecture-level cities decreases and the difference between REP indices increases.(3)The area of built-up areas, the area of green areas in built-up areas, the total amount of gas supply, and the length of sewerage pipes are the main limiting factors for cities other than Shanghai to improve their RES; the total amount of water resources, the area of sown seeds, the area of built-up areas, and road mileage are the main limiting factors for Shanghai to improve its RES.

## Figures and Tables

**Figure 1 ijerph-20-00240-f001:**
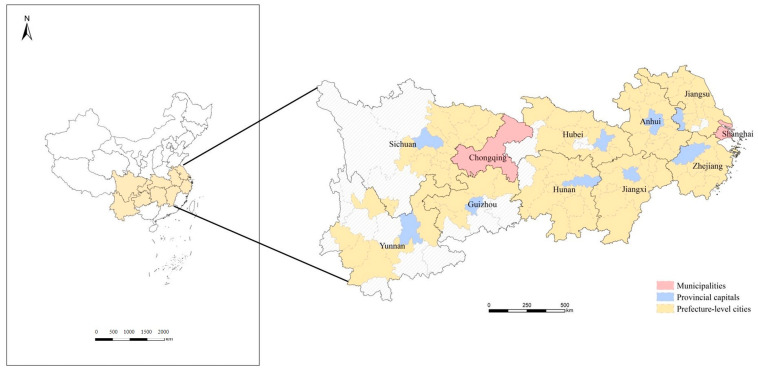
Spatial distribution of municipalities, provincial capitals, and prefecture-level cities in the Yangtze River Economic Belt (YREB).

**Figure 2 ijerph-20-00240-f002:**
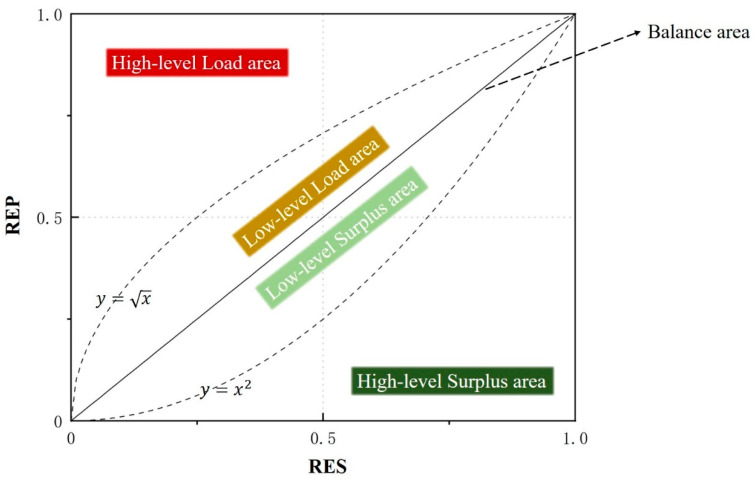
Coupling curve of RES and REP and partition states of RECC.

**Figure 3 ijerph-20-00240-f003:**
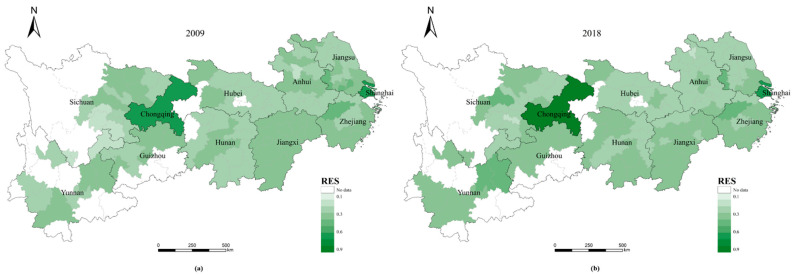
The spatial distribution of the RES index for cities in YREB in 2009 and 2018.

**Figure 4 ijerph-20-00240-f004:**
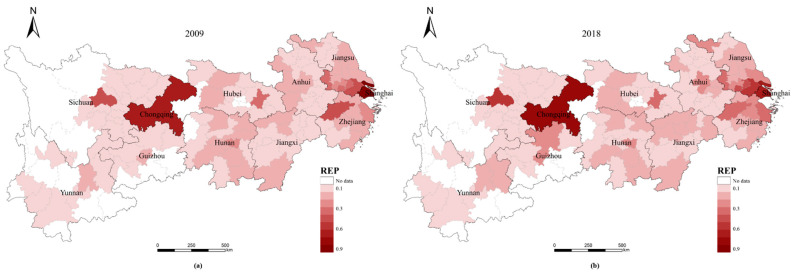
The spatial distribution of the REP index for cities in YREB in 2009 and 2018.

**Figure 5 ijerph-20-00240-f005:**
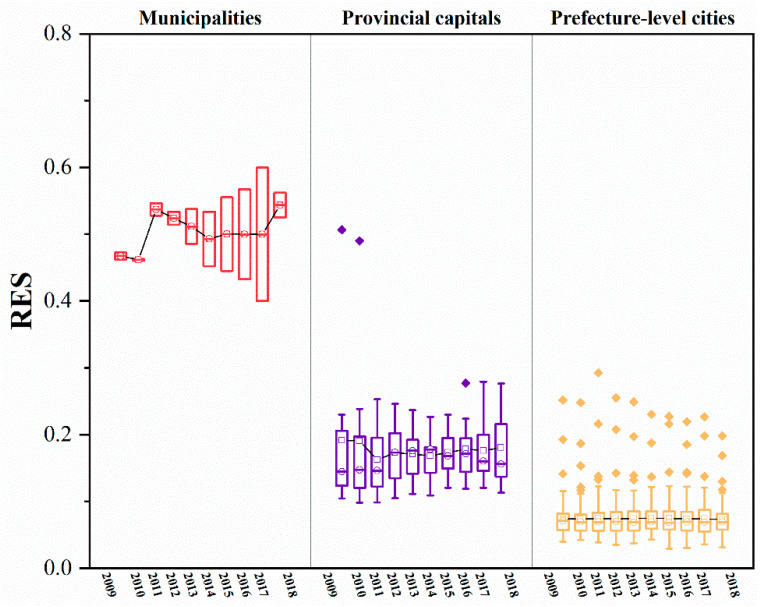
RES index of cities at different administrative levels in 2009 and 2018.

**Figure 6 ijerph-20-00240-f006:**
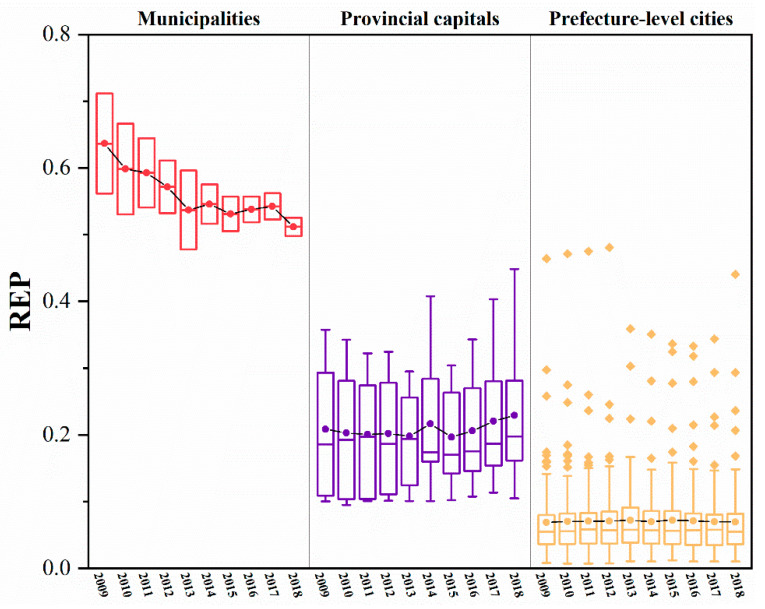
REP index of cities at different administrative levels in 2009 and 2018.

**Figure 7 ijerph-20-00240-f007:**
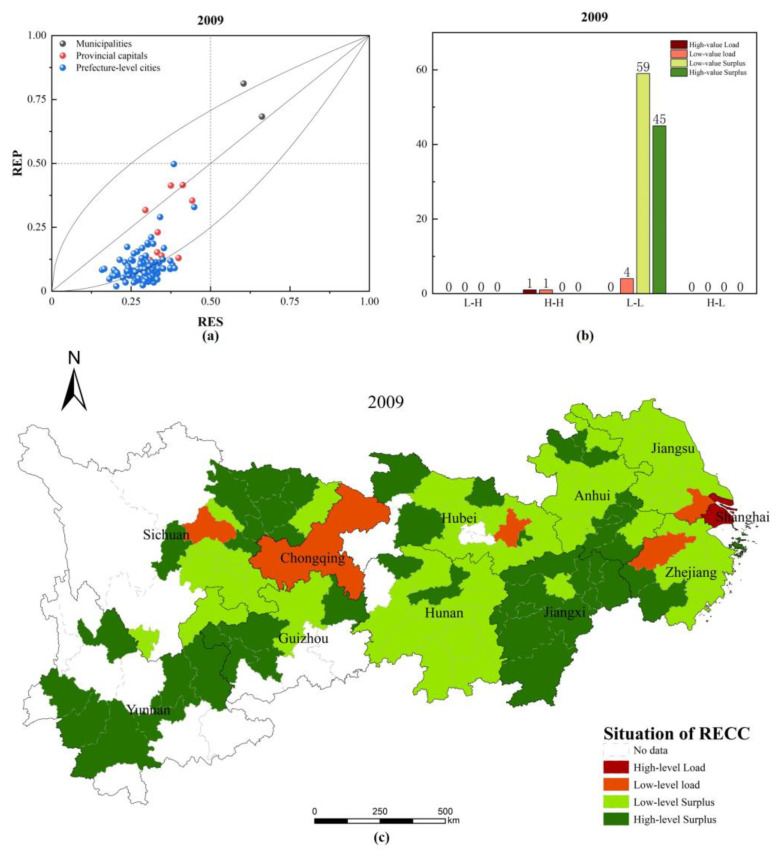
RECC status and city types of 110 cities in the Yangtze River Economic Belt in 2009 and 2018.

**Figure 8 ijerph-20-00240-f008:**
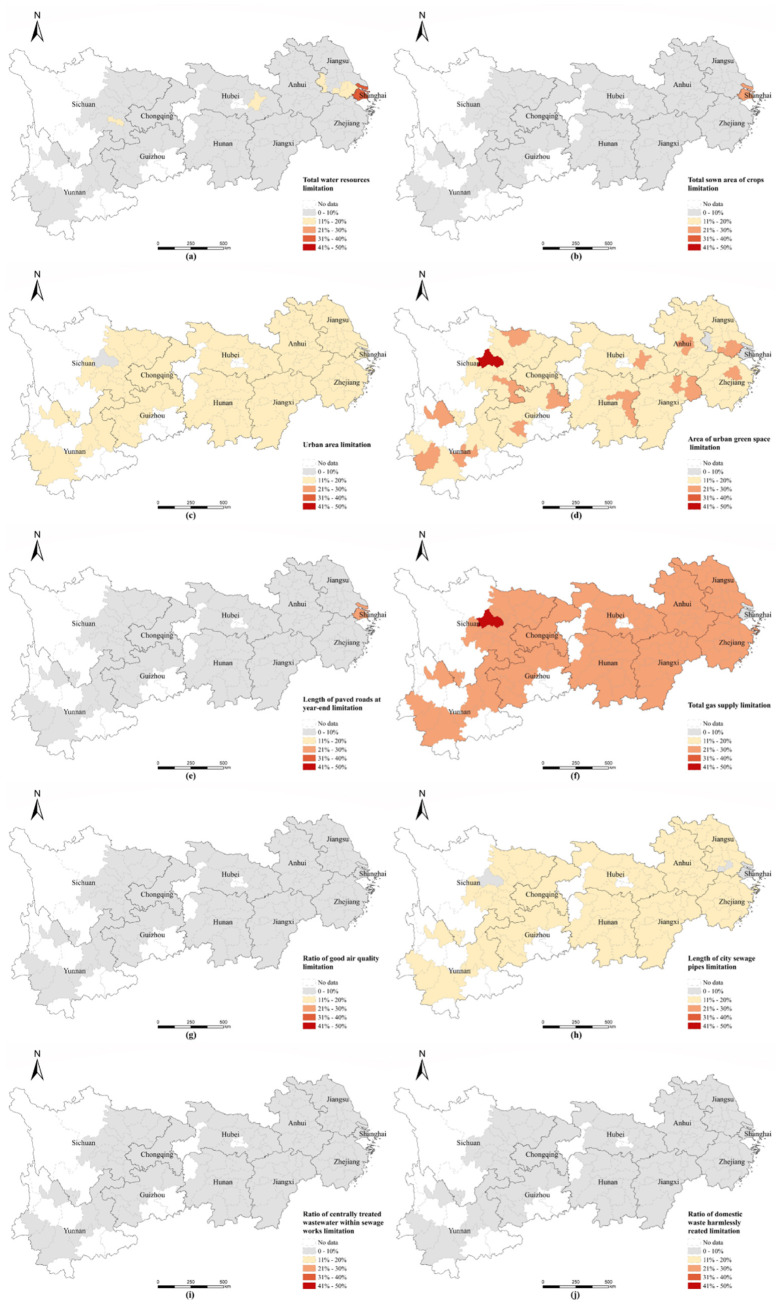
Obstacle factors and spatial distribution of RES index of cities in the Yangtze River Economic Belt.

**Table 1 ijerph-20-00240-t001:** Evaluation index system of RECC.

System	Criteria Layer	Indicators (Units)	System	Criteria Layer	Indicators (Units)
RES	Resources	S1 Total water resources (100 million tons)	REP	Economics	P1 GDP (CNY 100 million)
		S2 Grain sowing area (hectare)			P2 Industrial value-added (CNY 100 million)
		S3 Urban built-up area (km^2^)			P3 Total water consumption (100 million tons)
		S4 Green area of built-up area (hectare)			P4 Total electricity consumption (100 million kwh)
		S5 Highway mileage (km)			P5 Passenger traffic (10,000 persons)
		S6 Total gas supply (10,000 m^3^)			P6 Freight traffic (10,000 tons)
	Environment	S7 Ratio of good air quality (%)		Sociology	P7 Total population (10,000 persons)
		S8 Length of blowdown pipes (km)			P8 Urban unemployment rate (%)
		S9 Ratio of centrally treated wastewater within sewage works (%)			P9 Number of beds in hospitals and health centers (bed)
		S10 Ratio of domestic refuse disposal (%)		Pollution	P10 Industrial wastewater discharge (10,000 tons)

**Table 2 ijerph-20-00240-t002:** Classification of RECC status.

Balance	Low-Level Load Area	High-Level Load Area	Low-Level Surplus Area	High-Level Surplus Area
x = y	0 ≤x≤1	0≤x≤1	0≤x≤1	0≤x≤1
	$\sqrt{x}\leq y\leq 1$	$x\leq y<\sqrt{x}$	$x^{2}\leq y<x$	$0\leq y<x^{2}$

**Table 3 ijerph-20-00240-t003:** The coupling mechanism of the RES index and REP index.

	RES < 0.5	RES ≥ 0.5
REP ≥ 0.5	Low-High	High-High
REP < 0.5	Low-Low	High-Low

## Data Availability

The data presented in this study are openly available at https://data.cnki.net/ (accessed on 30 June 2021), and http://www.stats.gov.cn/ (accessed on 13 July 2021).

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
