# Peer review of "Evaluation of Resources and Environment Carrying Capacity Based on Support Pressure Coupling Mechanism: A Case Study of the Yangtze River Economic Belt"

_ijerph, 2022, doi:10.3390/ijerph20010240_

Round 1

Reviewer 1 Report (New Reviewer)

I think this is a good job. The topic and research work are very interesting. I just have some minor comments. On the section of introduction, the academic contribution of this paper to the research topic should be elaborated more clearly. I suggest the authors to add more texts to showcase the academic contribution of this paper.

Author Response

Thanks for suggestion. I have made some improvements as follows:

Point 1: The introduction is improved and academic contributions are more clearly described. As follows:

A number of limitations exist in the current stage of research on RECC.

(1) In existing studies, most scholars often divide the level of RECC index (usually a single quantitative value between 0 and 1) at equal intervals or directly use the grading standards of previous literature, which can only indicate the change in the in-tensity of RECC and cannot indicate the carrying state of RECC (i.e., overload state or surplus state). These subjective evaluation standards not only weaken the explanatory power and persuasiveness of the evaluation results to a certain extent, but also cannot determine whether the pressure of human activities on resources and environment exceeds the carrying capacity of these resources and environment, which is not conducive to an accurate understanding of the sustainable state of RECC.

(2) In existing studies [32,33], although RECC systems are usually divided into re-source, environmental and human activity layers, they are still eventually calculated as a whole, and the final results are unable to characterize the attributes of two relatively independent systems. In other words, it only calculates the support capacity of resources and environment and the overall pressure of human activities on these systems, without considering the coupling relationship between the two systems. It is not conducive to understanding the current state of these two systems and understanding the current state of these two systems or to further understanding the impact of re-source and environmental support activities and human activities on these systems.

(3) At present, there are few studies of the YREB RECC at the city level. Relevant studies have focused on specific regions [34-36]. However, the lack of research on RECC at the city level is not conducive to the establishment of a uniform early warning mechanism for RECC at the national level.

By combining human activities with resources and environment, the RECC evaluation system of bearing and pressure factors affecting bearing capacity is constructed, which is a supplement to existing research. The support index represents the potential of natural resources and social public resources created by human beings to promote development. The stress index represents the human economic and social activities served by the support system. Therefore, in this paper, the resource and environment carrying capacity index (RECC) is a single value in most other studies [37], which can be expressed as the ratio of the support index to the pressure index. Then the critical value of the overload (residual) area is determined according to the coupling mechanism between the support side and the pressure side, which is helpful to refine the RECC partition. In the field of social science, there are roughly two methods to measure the coupling between two or more systems: coupling degree model and coupling curve graph. The coupling degree model can carry out forward pressure on the coupling degree of support and pressure, but cannot fully reflect the state to RECC, so the critical value of RECC partition cannot be obtained. The coupling curve can express different coupling states, including the consideration of critical value, and can be transformed into expression mode through appropriate mathematical transformation to meet the needs of the research object. Therefore, this paper constructs a coupling curve diagram containing two variables, namely, the support side and the pressure side, to express the different coupling states of RECC, and obtains the critical value of RECC partition through the mathematical transformation of the coupling curve. Then, we can discuss the status of RECC according to the scale and support pressure coupling curve. It not only makes the evaluation results of RECC more convincing, but also better reflects whether RCCC is overloaded/surplus. Through the improved RECC index, we can scientifically determine the urban resource intensity and environmental carrying capacity, help us understand the changes in China's YREB urban RECC and urbanization process, and establish a RECC early warning mechanism.

Reviewer 2 Report (New Reviewer)

There is no doubt that this research paper is very interesting. It enriches the research on the carrying capacity of resources and environment. At the same time, this article has certain innovation in the construction of the indicator system of resource and environment carrying capacity. However, there are some weaknesses that require accurate inspection and review.

(1) There are too many format and language errors in the article, such as 26 lines, 231 lines (Where), 269 lines (An), 362 lines (b), etc. The formula in the text needs to be centered. In addition, the reference format needs to be modified according to the template requirements, such as Zhao, Y.; Cheng, J.; Zhu, Y.; Zhao, Y. Spatiotemporal Evolution and Regional Differences in the Production-Living-Ecological Space of the Urban Agglomeration in the Middle Reaches of the Yangtze River. Int. J. Environ. Res. Public Health 2021, 18, 59112497, doi:10.3390/ijerph182312497.

(2) The topic has clearly mentioned the analysis based on the coupling mechanism of support pressure, but its exact research idea cannot be known in the introduction. Therefore, the authors need to further elaborate the importance of this research perspective in the introduction.

(3) The paper should include more literature. Not enough relevant papers have been reviewed.

(4) The introduction needs to further clearly point out the value and significance of this research to the literature.

(5) In the discussion, the authors need to further emphasize the advantages of the coupling mechanism based on support pressure for analyzing the carrying capacity of resources and environment.

Author Response

Thanks for suggestion. I have made some improvements as follows:

Point 1: I have revised the format and language errors (including those you mentioned), centered the formula, and modified the format of references as required.

Point 2: The introduction has been improved, and the role of academic contributions and coupling mechanism has been more clearly described. As follows:

A number of limitations exist in the current stage of research on RECC.

(1) In existing studies, most scholars often divide the level of RECC index (usually a single quantitative value between 0 and 1) at equal intervals or directly use the grading standards of previous literature, which can only indicate the change in the in-tensity of RECC and cannot indicate the carrying state of RECC (i.e., overload state or surplus state). These subjective evaluation standards not only weaken the explanatory power and persuasiveness of the evaluation results to a certain extent, but also cannot determine whether the pressure of human activities on resources and environment exceeds the carrying capacity of these resources and environment, which is not conducive to an accurate understanding of the sustainable state of RECC.

(2) In existing studies [32,33], although RECC systems are usually divided into re-source, environmental and human activity layers, they are still eventually calculated as a whole, and the final results are unable to characterize the attributes of two relatively independent systems. In other words, it only calculates the support capacity of resources and environment and the overall pressure of human activities on these systems, without considering the coupling relationship between the two systems. It is not conducive to understanding the current state of these two systems and understanding the current state of these two systems or to further understanding the impact of re-source and environmental support activities and human activities on these systems.

(3) At present, there are few studies of the YREB RECC at the city level. Relevant studies have focused on specific regions [34-36]. However, the lack of research on RECC at the city level is not conducive to the establishment of a uniform early warning mechanism for RECC at the national level.

By combining human activities with resources and environment, this paper constructs a RECC evaluation system including support and pressure factors affecting bearing capacity, which complements the existing research. The support index represents the potential of natural resources and the social public resources created by hu-man beings to promote development. The pressure index represents the economic and social activities of human beings served by the support system. Therefore, in this paper, the resource and environment carrying capacity index (RECC), which is a single quantitative value in most other studies [37], can be expressed as the ratio of the support index and pressure index. Then this paper determines the critical value in the overload (surplus) area according to the coupling mechanism between the support side and the pressure side, which can help to refine the RECC partition. In the field of social science, there are roughly two methods to measure the coupling between two or more systems: coupling degree model and coupling curve graph. The coupling degree model can ex-press the coupling degree of support and pressure, but cannot comprehensively reflect the state to RECC, so the critical value of RECC partition cannot be obtained. The coupling curve chart can express different coupling states, and itself includes consideration of the critical value, and it can be transformed into an expression mode through appropriate mathematical transformation to meet the needs of the research object. Therefore, this paper constructs a coupling curve graph containing two variables, namely, the support side and the pressure side, to express the different coupling states of RECC, and obtains the critical value of RECC partition through the mathematical transformation of the coupling curve. Then, we can discuss the status of RECC according to the ratio and support pressure coupling curve. It not only makes the evaluation results of RECC more convincing, but also better reflects whether RCCC has overload/surplus. With this improved RECC index, we can scientifically determine the strength of urban resources and environmental carrying capacity, which helps us understand the changes in the RECC and urbanization process of the YREB cities in China, and to establish a RECC early warning mechanism.

Point 3: Add more references as follows:

  1. Li, J.; Cheng, J.; Wen, Y.; Cheng, J.; Ma, Z.; Hu, P.; Jiang, S. The Cause of China’s Haze Pollution: City Level Evidence Based on the Extended STIRPAT Model. IJERPH 2022, 19, 4597, doi:10.3390/ijerph19084597.
  2. Lv, A.; Han, Y.; Zhu, W.; Zhang, S.; Zhao, W. Risk Assessment of Water Resources Carrying Capacity in China. J. Am. Water Resour. Assoc. 2021, 57, 539–551, doi:10.1111/1752-1688.12936.

7      Han, C.; Lu, B.; Zheng, J. Analysis and Prediction of Land Resources’ Carrying Capacity in 31 Provinces of China from 2008 to 2016. Sustainability 2021, 13, 13383, doi:10.3390/su132313383.

  1. Zhou, W.; Elshkaki, A.; Zhong, S.; Shen, L. Study on Relative Carrying Capacity of Land Resources and Its Zoning in 31 Provinces of China. Sustainability 2021, 13, 1459, doi:10.3390/su13031459.

9      Li, B.; Zhang, H.; Huang, K.; He, G.; Guo, S.; Hou, R.; Zhang, P.; Wang, H.; Pan, H.; Fu, H.; et al. Regional Fauna-Flora Biodiversity and Conservation Strategy in China. iScience 2022, 25, 104897, doi:10.1016/j.isci.2022.104897.

  1. Tan, C.; Peng, Q.; Ding, T.; Zhou, Z. Regional Assessment of Land and Water Carrying Capacity and Utilization Efficiency in China. Sustainability 2021, 13, 9183, doi:10.3390/su13169183.
  2. Gao, Q.; Fang, C.; Cui, X. Carrying Capacity for SDGs: A Review of Connotation Evolution and Practice. Environ. Impact Assess. Rev. 2021, 91, 106676, doi:10.1016/j.eiar.2021.106676.
  3. Wei, C.; Dai, X.; Guo, Y.; Tong, X.; Wu, J. An Improved Approach of Integrated Carrying Capacity Prediction Based on TOPSIS-SPA. Sustainability 2022, 14, 4051, doi:10.3390/su14074051.
  4. Li, B.; Guan, M.; Zhan, L.; Liu, C.; Zhang, Z.; Jiang, H.; Zhang, Y.; Dong, G. Urban Comprehensive Carrying Capacity and Development Order: A “Pressure-Capacity-Potential” Logical Framework. Front. Environ. Sci. 2022, 10, 935498, doi:10.3389/fenvs.2022.935498.
  5. Luo, W.; Jin, C.; Shen, L. The Evolution of Land Resource Carrying Capacity in 35 Major Cities in China. Sustainability 2022, 14, 5178, doi:10.3390/su14095178.
  6. Yan, B.; Xu, Y. Evaluation and Prediction of Water Resources Carrying Capacity in Jiangsu Province, China. Water Policy 2022, 24, 324–344, doi:10.2166/wp.2022.172.
  7. Wang, H.; Cao, Y.; Wu, X.; Zhao, A.; Xie, Y. Estimation and Potential Analysis of Land Population Carrying Capacity in Shanghai Metropolis. Int. J. Environ. Res. Public Health 2022, 19, 8240, doi:10.3390/ijerph19148240.
  8. Ren, L.; Gao, J.; Song, S.; Li, Z.; Ni, J. Evaluation of Water Resources Carrying Capacity in Guiyang City. Water 2021, 13, 2155, doi:10.3390/w13162155.
  9. Huang, C.; Yin, K.; Liu, Z.; Cao, T. Spatial and Temporal Differences in the Green Efficiency of Water Resources in the Yangtze River Economic Belt and Their Influencing Factors. Int. J. Environ. Res. Public Health 2021, 18, 3101, doi:10.3390/ijerph18063101.
  10. Hu, J.; Ma, C.; Li, C. Can Green Innovation Improve Regional Environmental Carrying Capacity? An Empirical Analysis from China. Int. J. Environ. Res. Public Health 2022, 19, 13034, doi:10.3390/ijerph192013034.
  11. Tanveer, M., Khan, S.A.R., Umar, M. et al. Waste management and green technology: future trends in circular economy leading towards environmental sustainability. Environ Sci Pollut Res 2022,29, 80161–80178. Doi:10.1007/s11356-022-23238-8
  12. Khan, S.A.R.; Yu, Z.; Umar, M.; Tanveer, M. Green Capabilities and Green Purchasing Practices: A Strategy Striving towards Sustainable Operations. Business Strategy and the Environment 2022, 31, 1719–1729, doi:10.1002/bse.2979.

Point 4: In the discussion, the advantages of the coupling mechanism based on support pressure for analyzing the carrying capacity of resources and environment are further added. As follows:

In this paper, the RECC system is divided into two parts: the RES index, which reflects the resource security capacity and the environmental condition and governance level; and the REP index, which reflects the pressure of economic activities, social security and environmental pollution on the resource and environmental system. The indicators are weighted: the entropy weighting method is used to determine the weight coefficients of each indicator and the linear weighting method is used to calculate the RES index and REP index. In this paper, RECC index is the ratio of RES index and REP index, rather than the simple linear weighted sum (usually between 0 and 1). In addition, combined with the coupling curve, it can not only calculate the size of the RECC, but also judge the status of the RECC (overload/surplus), so as to more accurately formulate targeted development strategies for cities with different RECC statuses.

Reviewer 3 Report (New Reviewer)

They well written the article, to me they should have to add the literature. 

Must add more citations and references especially from the year 2021 and 2022. 

Few suggestions are to add but not limited to add 

 https://doi.org/10.1002/bse.2979

https://doi.org/10.1007/s11356-022-23238-8

Please add the future implications.

Also add a section of recommendations

Thanks 

Author Response

Thanks for suggestion. I have made some improvements as follows:

Point 1: Add more references (including those you suggest) as follows:

  1. Li, J.; Cheng, J.; Wen, Y.; Cheng, J.; Ma, Z.; Hu, P.; Jiang, S. The Cause of China’s Haze Pollution: City Level Evidence Based on the Extended STIRPAT Model. IJERPH 2022, 19, 4597, doi:10.3390/ijerph19084597.
  2. Lv, A.; Han, Y.; Zhu, W.; Zhang, S.; Zhao, W. Risk Assessment of Water Resources Carrying Capacity in China. J. Am. Water Resour. Assoc. 2021, 57, 539–551, doi:10.1111/1752-1688.12936.

7      Han, C.; Lu, B.; Zheng, J. Analysis and Prediction of Land Resources’ Carrying Capacity in 31 Provinces of China from 2008 to 2016. Sustainability 2021, 13, 13383, doi:10.3390/su132313383.

  1. Zhou, W.; Elshkaki, A.; Zhong, S.; Shen, L. Study on Relative Carrying Capacity of Land Resources and Its Zoning in 31 Provinces of China. Sustainability 2021, 13, 1459, doi:10.3390/su13031459.

9      Li, B.; Zhang, H.; Huang, K.; He, G.; Guo, S.; Hou, R.; Zhang, P.; Wang, H.; Pan, H.; Fu, H.; et al. Regional Fauna-Flora Biodiversity and Conservation Strategy in China. iScience 2022, 25, 104897, doi:10.1016/j.isci.2022.104897.

  1. Tan, C.; Peng, Q.; Ding, T.; Zhou, Z. Regional Assessment of Land and Water Carrying Capacity and Utilization Efficiency in China. Sustainability 2021, 13, 9183, doi:10.3390/su13169183.
  2. Gao, Q.; Fang, C.; Cui, X. Carrying Capacity for SDGs: A Review of Connotation Evolution and Practice. Environ. Impact Assess. Rev. 2021, 91, 106676, doi:10.1016/j.eiar.2021.106676.
  3. Wei, C.; Dai, X.; Guo, Y.; Tong, X.; Wu, J. An Improved Approach of Integrated Carrying Capacity Prediction Based on TOPSIS-SPA. Sustainability 2022, 14, 4051, doi:10.3390/su14074051.
  4. Li, B.; Guan, M.; Zhan, L.; Liu, C.; Zhang, Z.; Jiang, H.; Zhang, Y.; Dong, G. Urban Comprehensive Carrying Capacity and Development Order: A “Pressure-Capacity-Potential” Logical Framework. Front. Environ. Sci. 2022, 10, 935498, doi:10.3389/fenvs.2022.935498.
  5. Luo, W.; Jin, C.; Shen, L. The Evolution of Land Resource Carrying Capacity in 35 Major Cities in China. Sustainability 2022, 14, 5178, doi:10.3390/su14095178.
  6. Yan, B.; Xu, Y. Evaluation and Prediction of Water Resources Carrying Capacity in Jiangsu Province, China. Water Policy 2022, 24, 324–344, doi:10.2166/wp.2022.172.
  7. Wang, H.; Cao, Y.; Wu, X.; Zhao, A.; Xie, Y. Estimation and Potential Analysis of Land Population Carrying Capacity in Shanghai Metropolis. Int. J. Environ. Res. Public Health 2022, 19, 8240, doi:10.3390/ijerph19148240.
  8. Ren, L.; Gao, J.; Song, S.; Li, Z.; Ni, J. Evaluation of Water Resources Carrying Capacity in Guiyang City. Water 2021, 13, 2155, doi:10.3390/w13162155.
  9. Huang, C.; Yin, K.; Liu, Z.; Cao, T. Spatial and Temporal Differences in the Green Efficiency of Water Resources in the Yangtze River Economic Belt and Their Influencing Factors. Int. J. Environ. Res. Public Health 2021, 18, 3101, doi:10.3390/ijerph18063101.
  10. Hu, J.; Ma, C.; Li, C. Can Green Innovation Improve Regional Environmental Carrying Capacity? An Empirical Analysis from China. Int. J. Environ. Res. Public Health 2022, 19, 13034, doi:10.3390/ijerph192013034.
  11. Tanveer, M., Khan, S.A.R., Umar, M. et al. Waste management and green technology: future trends in circular economy leading towards environmental sustainability. Environ Sci Pollut Res 2022,29, 80161–80178. Doi:10.1007/s11356-022-23238-8
  12. Khan, S.A.R.; Yu, Z.; Umar, M.; Tanveer, M. Green Capabilities and Green Purchasing Practices: A Strategy Striving towards Sustainable Operations. Business Strategy and the Environment 2022, 31, 1719–1729, doi:10.1002/bse.2979.

Point 2: The the future implications and recommendations are in lines 540 to 570. As follows:

Combined with the above analysis, the following policy suggestions are formu-lated to provide a relevant basis for the sustainable development of the YREB.

For municipal authorities and provincial capitals, their economic development level has reached a high level, so the relationship between human activities and re-sources and environmental systems in these cities is extremely tense. These locations should implement strict water resource management and rational land use planning in the future, and large-scale resource redeployment projects across regions such as south-north water transfer and west-east gas transmission still need to be built. In ad-dition, these cities should increase the supply of social resources such as urban green space and urban road areas. To solve the problem of resource scarcity, cities must change their production methods and lifestyles to improve the efficiency of resource use. Advanced cities should improve green innovation capabilities [50], develop waste management and green technologies, such as domestic garbage and water resources recycling technology [51] and improve green capabilities to achieve sustainable opera-tion [52].

For the prefecture-level cities, their economic development is not good enough, which makes them have a strong need for economic development. Therefore they have to make development plans in advance, prioritize the development of resource-saving and environment-friendly industries, and focus on the development of the circular economy. The government should strengthen investment in environmental manage-ment and improve the coverage of technologies such as water purification technology, industrial sulfur dioxide treatment technology, industrial wastewater treatment tech-nology and domestic waste treatment technology to effectively reduce sewage dis-charge and waste generation. The city should pay attention to reducing the impact on and damage to existing resources and the environment while developing economically and actively exploring the path of sustainable development.

This paper evaluates only some of the cities’ resources, environment and econom-ic and social activities and does not address trade and other mobility factors between cities and regions. In addition, arable land area, forest area, food production, energy consumption and some other factors were not considered due to problems in data col-lection. The driving forces behind changes in RECC in cities have not been studied and need to be assessed in future studies.

This manuscript is a resubmission of an earlier submission. The following is a list of the peer review reports and author responses from that submission.

Round 1

Reviewer 1 Report

The manuscript conducts statistical analysis based on yearbook data. Results and conclusions are well arranged, however, the study still has some major flaws.

(1) Author affiliations are in Chinese?

(2) Missing 'space' after Keywords.

(3) Conclusion is the conclusion, such recommendations should be put before this section and may be appropriate just after the discussions.

 (4) Very poor quality of all figures (low resolution, content too small to read)

(5) Missing article info (no funding info, no references)

(6) No reference cited in the Discussion

Thus, I would recommend rejecting the manuscript according to above major flaws.

Reviewer 2 Report

To authors:

In the manuscript, the authors have discussed how the RECC concept can be applied to the study area in Yangtze REB. 

  • The authors have included details of methods, results, and discussions in their manuscript.
  • The authors highlighted the merit and applicability of the given methods and application examples.

However, I also suggest improving your manuscript by adding/revising/editing following

1) The manuscript does not provide any references, which is a critical shortfall.  Without references and appropriate citations in the manuscript, the decision cannot be made. 

2) The method discussed has little merit or contribution in terms of academic discussion.

3) Most maps (for example future 8) are too small to verify the results. They should be included in a high resolution

4) The manuscript should include in-depth discussion/literature regarding the AHP-CV model and how they are applied in similar studies discussed in this manuscript. 

Overall, I believe the manuscript only can be considered after the major revision which would provide references and an in-depth literature review. 

Reviewer 3 Report

About your article, entitled : "Evaluation of resources and environment carrying capacity based on support pressure coupling mechanism: a case study of the Yangtze River Economic Belt", i  think that excellent work has been done. I think the data and methods section is very well structured. The scientific literature used, I think, is adequate. The results and conclusions section is in my view, very well structured. I find the Policy recommendations section very interesting for readers.

Best Regards!

Author Response

Thanks very much for taking your time to review this manuscript. Thank you for your approval! I hope everything goes well with you!